# Connectivity in Large-Scale Resting-State Brain Networks Is Related to Motor Learning: A High-Density EEG Study

**DOI:** 10.3390/brainsci12050530

**Published:** 2022-04-21

**Authors:** Simon Titone, Jessica Samogin, Philippe Peigneux, Stephan Swinnen, Dante Mantini, Genevieve Albouy

**Affiliations:** 1Movement Control and Neuroplasticity Research Group, Department of Movement Sciences, KU Leuven, 3001 Leuven, Belgium; simon.titone@kuleuven.be (S.T.); jessica.samogin@kuleuven.be (J.S.); stephan.swinnen@kuleuven.be (S.S.); dante.mantini@kuleuven.be (D.M.); 2LBI–KU Leuven Brain Institute, KU Leuven, 3001 Leuven, Belgium; 3Neuropsychology and Functional Neuroimaging Research Group (UR2NF), Centre for Research in Cognition and Neurosciences (CRCN), Université Libre de Bruxelles (ULB), 1050 Brussels, Belgium; philippe.peigneux@ulb.be; 4Department of Health and Kinesiology, College of Health, University of Utah, Salt Lake City, UT 84112, USA

**Keywords:** resting-state functional connectivity, brain networks, motor learning, high density electroencephalography, sleep, motor memory consolidation

## Abstract

Previous research has shown that resting-state functional connectivity (rsFC) between different brain regions (seeds) is related to motor learning and motor memory consolidation. Using high-density electroencephalography (hdEEG), we addressed this question from a brain network perspective. Specifically, we examined frequency-dependent functional connectivity in resting-state networks from twenty-nine young healthy participants before and after they were trained on a motor sequence learning task. Consolidation was assessed with an overnight retest on the motor task. Our results showed training-related decreases in gamma-band connectivity within the motor network, and between the motor and functionally distinct resting-state networks including the attentional network. Brain-behavior correlation analyses revealed that baseline beta, delta, and theta rsFC were related to subsequent motor learning and memory consolidation such that lower connectivity within the motor network and between the motor and several distinct resting-state networks was correlated with better learning and overnight consolidation. Lastly, training-related increases in beta-band connectivity between the motor and the visual networks were related to greater consolidation. Altogether, our results indicate that connectivity in large-scale resting-state brain networks is related to—and modulated by—motor learning and memory consolidation processes. These finding corroborate previous seed-based connectivity research and provide evidence that frequency-dependent functional connectivity in resting-state networks is critically linked to motor learning and memory consolidation.

## 1. Introduction

The acquisition and retention of new motor behaviors (e.g., typing, knitting, or riding a bicycle) is integral to everyday life. Motor learning can be described as the increased spatial and temporal accuracy of such new sets of movements with practice [1]. In order for these improvements to persist in the long term, consolidation must occur [2]. Numerous empirical studies in the past decades have provided evidence that sleep benefits motor memory consolidation at the behavioral level [3,4,5]. The neural activity patterns supporting motor learning and (sleep-related) motor memory consolidation have been well described using functional magnetic resonance imaging (fMRI) [6,7,8]. However, brain connectivity patterns associated to these processes have only more recently been investigated. Specifically, a large body of fMRI studies has highlighted the importance of studying resting-state functional connectivity (rsFC) in the context of motor processes. Previous research has indeed shown that motor learning can influence rsFC patterns [9], but also that motor performance [10] and motor memory consolidation [11] are linked to the strength of rsFC within or between specific networks.

While fMRI has traditionally been the primary research technique used to examine rsFC, recent methodological developments [12] have allowed such investigations to be performed using electrophysiological techniques such as high-density electroencephalography (hdEEG) and magnetoencephalography (MEG). In particular, it has been shown that both hdEEG and MEG can be used to examine rsFC at the source level [13,14]. These methodologies allow for a temporally fine-grained investigation into functional interactions between seeds and/or large-scale brain networks, and allow direct measurements of oscillatory activity in different frequency bands [13]. Using seed-based source-localized rsFC approaches, recent MEG studies have provided additional insights into the relationship between rsFC and motor behavior. Specifically, it has been shown that lower baseline (pre-task) connectivity between anatomically distant seeds (i.e., between the somatomotor cortex and both the putamen and cerebellum) in the beta frequency band is related to better learning of a finger-tapping task [15,16]. Additionally, greater training-related changes in beta-band rsFC between motor seeds (i.e., right primary sensorimotor -rSM1- and pre-SMA) were related to better learning during the initial training phase [17]. Interestingly, training-related changes in beta-band connectivity were also linked to the consolidation process. Specifically, post-task increases in beta-band rsFC between rSM1 and frontal as well as parietal areas were described to be correlated with a motor performance boost observed during an early consolidation window [17].

Importantly, the studies reviewed above mainly used *seed*-*based* connectivity approaches, whereas the investigation of connectivity patterns from a *network* perspective in the context of motor learning is currently limited. This is surprising as neuroimaging research has highlighted the value of studying patterns of connectivity between multiple regions spatially distributed in functional large-scale networks [18,19]. Such functional network patterns are thought to reflect the synchrony of neuronal populations involved in a common function and wired together through plasticity [20]. Accordingly, examining connectivity at a network level can bring additional insights into plasticity processes underlying motor learning and control [10]. The goal of the present research was, therefore, to use hdEEG to examine source-localized rsFC patterns at the *network level* and to test whether such connectivity patterns are modulated by task practice, and/or related to the learning and sleep-related consolidation processes. To do so, resting-state hdEEG data were acquired from young healthy individuals immediately before and after they were trained on a motor sequence learning (MSL) task. Consolidation was assessed with an overnight retest on the motor task. We examined (i) the effect of learning on functional connectivity within and between resting-state networks previously described in earlier fMRI [21] and hdEEG [12] research, and (ii) the link between these connectivity patterns and behavioral indices of initial (online) motor learning and sleep-related (offline) motor memory consolidation. Since oscillatory activity in the beta [15,16,17], alpha [22], and gamma [23] bands has been preferentially associated to motor learning and control processes in earlier research, these frequency bands were considered of primary interest in the present study.

We expected motor learning to induce changes in alpha, beta, and gamma-band connectivity within the motor network, and between motor and functionally distinct networks [10,17,24]. Based on previous seed-based connectivity research linking beta-band [15,17], but not alpha-band [16] connectivity patterns to motor performance, we predicted that our network approach would preferentially highlight relationships between large-scale connectivity in the beta band and motor performance. To the best of our knowledge, this is the first study to examine the relationship between gamma-band rsFC at the network level and motor performance. However, based on earlier EEG research reporting modulation of gamma connectivity as a function of performance improvement on a motor task [25], we hypothesized that the strength of the gamma-band connectivity at rest would also be related to motor performance.

## 2. Materials and Methods

### 2.1. Participants

Twenty-nine young (23.7 ± 3.3 years old, 15 females) right-handed [26] healthy volunteers were recruited to complete this study. Participants had no history of medical, neurological, or psychiatric disorders. They also showed no evidence of anxiety [27], depression [28] or excessive daytime sleepiness [29]. Participants were not experienced video gamers, trained musicians, nor typists. Further, they had normal sleep quality as assessed by the Pittsburgh Sleep Quality Index questionnaire [30] and the Saint Mary’s Hospital questionnaire [31]. None of the participants were extreme morning or evening chronotypes [32] or shift-workers, nor did any report trans-meridian trips in the three months prior to the experiment.

Out of the 29 participants who took part in the study, 2 participants were excluded from the analyses due to an experimental error that resulted in additional task practice. The remaining 27 participants (23.6 ± 3.5 y/o, 13 females; see Appendix A for participant demographics) were included in the analyses. Of these 27 participants, behavioral data of specific sessions were excluded from the behavioral analyses due to technical issues that resulted in inaccurate recording of motor performance. Specifically, behavioral training and retest data were excluded for three participants, offline gain data for three other participants, and retest data for one participant. Behavioral data analyses, therefore, included 24 participants for training, 21 participants for offline changes in performance, and 20 participants for retest. Additionally, of the 27 participants included in the analysis, EEG data of specific sessions were excluded from the EEG analyses. Specifically, EEG data of one session (RS pre-task) were missing for two participants, EEG data were of bad quality for two participants, and two additional participants were excluded from the EEG analyses as they were statistical outliers (mean ± 3 × standard deviation) in connectivity metrics across multiple frequency bands. Eventually, 21 participants were included in the EEG analyses testing for session effects, 18 in the EEG analyses using (training) online changes in performance as covariate, and 15 in the EEG analyses using offline changes in performance as covariate. The different sample sizes are reported in Table 1 below and in the results section for each analysis.

To test whether the exclusion of participants did not compromise the power of our brain-behavior correlation analyses, we performed a power analysis using G*Power software [33] based on effect sizes reported in similar earlier research [15,17]. For the correlation between rsFC and online gains in performance, we used the r-values reported in [15], corresponding to the correlation using changes in performance during initial motor sequence learning as dependent variable and baseline rsFC as independent variable. The effect size was 0.87 and a power of 96.1% was reached with 10 participants at an alpha error probability of 0.05. Our sample size for the corresponding brain-behavior analyses (online gains) is of 18 participants. For the correlation with offline gains, we used the r-values reported in [17], corresponding to the correlation using offline changes in motor performance in an early consolidation window as dependent variable and rsFC as independent variable. The effect size was 0.83, and a power of 96.1% was reached with 12 participants at an alpha error probability of 0.05. Our sample size for the corresponding brain-behavior analyses (offline gains) is of 15 participants.

### 2.2. Experimental Methods

The research presented in this manuscript is part of a larger experimental protocol completed by the same cohort of participants (see Appendix A for the full design).

One to four weeks prior to the experimental session, participants were invited to the sleep lab for a habituation night, during which they were introduced to the general procedure of the study, and slept overnight with the hdEEG cap on (Figure 1). They also wore additional electrodes on the legs, and a respiratory belt and thermistor to assess possible sleep disorders. Four days prior to the experimental session, participants were asked to follow a constant sleep schedule (according to their own rhythm ± 1 h), and to refrain from drinking alcohol and taking naps. Compliance to the requested schedule was assessed using sleep diaries and wrist actigraphy measures (ActiGraph wGT3X-BT, Pensacola, FL, USA). Participants were invited to spend 12 consecutive hours in the laboratory overnight (from 7–9 pm on day 1 to 7–9 am on day 2). On day 1, after receiving a general overview of the study and preparing for the night, participants were equipped with the hdEEG system. EEG data were recorded during a five-minute resting-state (RS) period prior to task performance, referred to as RS pre-task (start RS pre-task: 9–11 pm). During the RS session, participants sat upright in a chair and were instructed to sit still with eyes closed, to try not to think about anything in particular, and to let their mind wander. General motor execution skill was then assessed using a random Serial Reaction Time (SRT) task [34] (see Appendix A). Participants were then trained on the motor sequence learning task (start training between 9–11 pm) while being monitored with hdEEG (data not reported in the present manuscript). Immediately after training ended, a five-minute RS period monitored with hdEEG and referred to as RS post-task was acquired with the same instructions as described above. After the motor learning task and resting-state session, participants were free to read or watch TV until they felt ready to sleep (bedtime between 10:15 pm–12:30 am). Before lights were turned off for the night, a 3D scan of the electrodes positions was acquired. The next morning (wake time between 6:30–8:15 am; note that one participant naturally woke up at 5:40 am—thus before the earliest lights on time of 6:30 am—and was therefore retested at 6:25 am on the task—thus before the earliest task retest time of 7:15 am. Importantly, this participant presented 7 h of sleep and reported to be fully rested during the morning retest (SSS: 1)), participants had a second 3D scan, and the EEG cap was removed before participants were retested on the MSL task (retest session). This was done at least 45 min after wake time to allow sleep inertia to dissipate (start MSL retest: 7:15–9:15 am). Participants were also invited on a separate day to the MR facility to acquire an anatomical T1 scan, which was used for EEG data processing (see below).

Vigilance was assessed prior to MSL training and retest objectively using a psychomotor vigilance test (PVT) [35], and subjectively using the Stanford Sleepiness Scale (SSS) [36].

### 2.3. Motor Sequence Learning Task

Participants performed an explicit bimanual finger-tapping task implemented in MATLAB with Psychophysics Toolbox version 3 [37] both before sleeping (training) and 45 min after waking the following morning (retest). The task was completed while sitting upright. During the task, participants used both hands (excluding the thumbs) to tap a six-element finger sequence on a specialized keyboard as quickly and accurately as possible. The learned sequence (4-7-3-8-2-6, where 1 and 8 correspond to the left and right pinkie fingers, respectively) was memorized before the start of training. Two pre-training runs were performed before the training and retest sessions to ensure that participants were able to memorize the sequence. In pre-training 1, the sequence of numbers corresponding to the fingers was displayed on the screen while participants completed the sequence until three consecutive accurate sequences were recorded. During pre-training 2, the sequence was not visible but replaced by a green cross in the middle of the screen, and participants again practiced the task until three consecutive accurate sequences were recorded.

The MSL training and retest sessions each consisted of 20 blocks, and all participants were trained on the same sequence, during both sessions. In each session, participants were first trained with 16 blocks of practice, and their performance was subsequently tested with 4 blocks of practice administered after a short break of approximately one minute in order to reduce the effect of fatigue on end-training performance [38]. The start of each practice block was indicated by a green fixation cross in the center of the screen, and participants were instructed to type the sequence as quickly and as accurately as possible until a red cross appeared. Practice blocks consisted of 60 key presses for a total of 10 possible repetitions of the sequence. Rest periods (duration 20 s) were offered to the participants after each practice block. During rest blocks, a red cross was visible on the screen, and participants were instructed to keep their fingers still on the keys and to continue fixating on the cross. In the last 5 s of each rest block, the sequence was displayed as a reminder before starting the next practice block.

Motor performance was quantified with measures of speed (mean time to complete an accurate transition (in s) and accuracy (% of correct transitions). The analyses of these behavioral data were performed with SPSS Statistics 24 (IBM) using repeated-measures ANOVA. Online changes in performance, defined as the difference between averaged performance across the first 4 blocks and the last 4 blocks of training, were measured to assess initial learning-related changes in performance. Offline changes in performance, defined as the difference between averaged performance across the last 4 (fatigue-free) blocks of training and the first 4 blocks of retest, were measured to assess consolidation-related changes in performance. Separate t tests were performed to test whether online and offline changes in performance were different from zero. Note that online and offline changes in performance speed were used as covariates in the EEG analyses examining brain-behavior relationships.

### 2.4. Neuroimaging Data Acquisition

#### 2.4.1. EEG Data Acquisition

Brain activity during the RS sessions, the MSL training session (data not reported here) as well as the nights of sleep (habituation and experimental nights) were recorded using 128 channel hdEEG ActiCHamp and BrainAmp amplifiers from BrainProducts (ActiCHamp and BrainAmp, Brain Products GmbH, Gilching, Germany) at a sampling rate of 1000 Hz, using electrode FCz as recording reference. Bilateral horizontal and vertical electrooculogram (EOG), along with chin and trapezius electromyogram (EMG) were recorded simultaneously with EEG recordings. Positions of EEG sensors were recorded using a structure sensor camera and Skanect 3D software from Occipital Inc. (Boulder, CO, USA), and localized using the SPOT3D toolbox [39].

#### 2.4.2. MRI Data Acquisition

An anatomical T1-weighted 3D MP-RAGE sequence (TR = 9.5 ms; TE = 4.6 ms; TI = 858.1 ms; FA = 9°; 160 slices; FoV = 250 × 250 mm^2^; matrix size = 256 × 256 × 160; voxel size = 0.98 × 0.98 × 1.20 mm^3^) was acquired using a Phillips Achieva 3.0 T MRI System with a 32-channel head coil. RS fMRI data (not reported in the present manuscript) were also acquired with an ascending gradient EPI pulse sequence for T2*-weighted images (TR/TE = 1000/33 ms; multiband factor 3; flip angle = 70°; 42 transverse slices; interslice gap = 0.5 mm; voxel size = 2.14 × 2.18 × 3 mm^3^; FoV = 240 × 240 × 146.5 mm^3^; matrix = 112 × 110; 600 dynamic scans). During the RS scan (~10 min), participants were instructed to keep their eyes closed and to not think of anything in particular.

### 2.5. Neuroimaging Data Analysis

#### 2.5.1. Sleep Scoring

Sleep was scored using an automated algorithm [40] that uses two EEG (C3 and C4, mastoid reference) and EOG (left and right EOG, mastoid reference) channels. The two EEG channels were averaged, and band-pass filtered between 0.3–45 Hz while the EOG was filtered between 0.3–12 Hz. Data were then downsampled to 100 Hz and divided into 30 s epochs. Short-time Fourier transform with a Hamming window length of 128 samples was then performed. A sixteen-layer deep convolutional neural network used the resulting spectrogram to assign a class probability to each epoch, with the sleep stage having the highest probability being considered the most probable class (MPC). A multilayer perceptron then weighed the MPC outputs to generate a revised class probability for the stage of the epoch, and the MPC was again calculated from that revised score. The automated sleep scores were visually checked epoch by epoch by an experimenter (ST). In case of a mismatch between the experimenter’s and the algorithm’s score, the experimenter’s score was retained for the epoch. Sleep scoring outputs (e.g., total sleep time, sleep onset latency, sleep efficiency, etc.) reported in Appendix A were used to objectively assess sleep quality and quantity during the post-learning night spent in the lab.

#### 2.5.2. hdEEG Resting-State Functional Connectivity Analyses

Frequency dependent functional connectivity was derived from the resting-state EEG data with an automated workflow developed and validated by our team [12,41].

#### 2.5.3. EEG Signals Preprocessing

EEG data preprocessing included the correction of “bad channels” and the attenuation of noise as well as biological artifacts [13,41]. We first detected channels with low signal quality and labeled them as “bad channels” using the procedures described in Samogin et al., 2020 [12]. Next, the signal of each “bad channel” was reconstructed from spatial interpolation of the neighboring channels, defined using the FieldTrip toolbox (http://www.fieldtriptoolbox.org (accessed on 1 February 2021)). The resulting EEG data was band-pass filtered (1–80 Hz) using EEGLab (https://sccn.ucsd.edu/eeglab (accessed on 1 February 2021)), and subsequently re-referenced to an average reference by removing the mean value across channels [42]. Independent component analysis (ICA) was then used to attenuate ocular and muscular artifacts in the EEG signals [43]. The artifact detection solution implemented in Liu et al., 2017 [13] was used to automatically identify ICs associated with artifacts. Non-artifactual ICs were mixed linearly with the ICA-estimated weights in order to obtain the artifact-corrected EEG signals [12].

#### 2.5.4. Individual Head Model Creation

The T1 anatomical MR image was segmented into 12 different classes of tissue. A realistic head model was then reconstructed from the segmented T1 and the EEG sensor positions, which were rigidly co-registered to the contour of the head [13,41]. A 3D regular 6 mm grid was overlapped on layers corresponding to cortical, subcortical, and cerebellar gray matter to define all possible dipole sources. Conductivity values were defined based on previous literature [44]. Next, SimBio [45,46] was used to generate the whole-head finite element head model. Finally, the head model was used to calculate a leadfield matrix expressing the linear relationship between scalp EEG data and source-space neural activity.

#### 2.5.5. EEG Source Reconstruction

The exact low-resolution brain electromagnetic tomography (eLORETA) algorithm was used to compute the cortical 3-D distribution of current density derived from processed scalp EEG data [47]. The algorithm estimated source-space neural activity in a 6 mm homogenous grid constrained to the gray matter, using the leadfield matrix as input.

#### 2.5.6. Network Connectivity Analyses

This research examined connectivity within and between six resting-state networks (RSNs) that have been commonly investigated in other studies. These networks cover a large portion of the cortical surface, and include the default mode network (DMN), dorsal attention network (DAN), ventral attention network (VAN), somatomotor network (MOT), visual network (VIS), and language network (LANG). Based on previous literature on motor learning and control, connectivity within the MOT network [48] as well as between the MOT network and all other networks [10], is of interest in the present study. Note that, for the sake of completeness, connectivity analyses within and between the other networks described above are reported in the current paper but are considered exploratory. The nodes of these RSNs were selected based on previous literature [12,21] that established canonical rsFC networks (Table 2), and have been most commonly investigated in previous studies using different modalities including fMRI [21,49,50], hdEEG [12,41], and MEG [16,17]. A minimum distance of 15 mm was imposed between ROIs to minimize spurious effects of signal leakage (see Table 2 for coordinates).

The 21 chosen regions of interest (ROI) were defined in the Montreal National Institute (MNI) space, and then projected to individual space. For each ROI, a sphere with a 6 mm radius was defined around the MNI coordinate. For each resulting spherical ROI, time-courses of gray-matter voxels were decomposed in the time-frequency domain using the short-time Fourier transform. EEG connectivity was then calculated using power envelope correlations between orthogonalized signals [51]. Specifically, Pearson correlations were computed between ROIs, per frequency band (range: 1–80 Hz) on the logarithmic-transformed signal-orthogonalized power time-courses. The resulting r-values were then transformed to z-scores using the Fisher transform [21,51]. Next, z-scores were averaged within relevant frequencies to extract frequency band connectivity values between specific ROIs for the delta (1–4 Hz), theta (4–8 Hz), alpha (8–13 Hz), beta (13–30 Hz), and gamma (30–80 Hz) bands. Based on previous literature on motor learning and control, frequency bands of interest in the current study included the alpha [16], beta [15], and gamma [23] bands. Analyses of the other frequency bands (i.e., delta and theta) are reported in the present paper for the sake of completeness but are considered as exploratory.

RSN connectivity was examined at both the within-network and the between-network levels. Within-network connectivity was computed as the averaged connectivity between pairs of ROIs within each network. In contrast, between-network connectivity was calculated as the averaged connectivity between all possible pairs of ROIs from two distinct networks.

Statistical analyses were performed in MATLAB 2016b on the resulting within and between-network connectivity values, separately for the 5 frequency bands described above. First, we tested whether rsFC changed from the pre- to the post-task sessions using a repeated measures ANOVA. Next, we examined whether connectivity metrics were correlated with behavioral indices (i.e., online and offline changes in performance). Specifically, using Pearson’s correlation, we tested for relationships between the behavioral indices and (1) pre-task (baseline) rsFC and (2) the change in rsFC between pre- and post-task sessions. Exploratory analyses testing for correlations between post-task rsFC and behavioral measures are reported in the Appendix A. For each of these analyses, the false discovery rate (FDR) method [52] was used to correct for multiple comparisons. Correction was applied across the pairs of networks of interest (i.e., 6 pairs of interest: MOT-MOT, MOT-DMN, MOT-DAN, MOT-VAN, MOT-VIS, MOT-LANG). We used a more stringent correction for the exploratory analyses including all other networks pairs and corrected for all pairs tested (i.e., 21 pairs). Uncorrected p-values are reported in the main text, and the figures indicate whether these effects survived FDR correction for multiple comparisons.

## 3. Results

### 3.1. Behavior

#### 3.1.1. Sleep and Vigilance

Actiwatch and sleep diary data indicated that participants slept regularly and more than 7 h per night during the 4 nights prior to the experimental night (mean sleep duration across the 4 nights: 8.28 h ± 0.12 min; see details in Appendix A). During the overnight sleep recording in the lab, average total sleep time was 7 h 17 min ± 1 h 6 min, and sleep efficiency was 93.6% ± 11.3% (see Appendix A). Subjective measures of vigilance (SSS) indicated that participants were more alert during the morning retest as compared to the nighttime training, but such differences were not observed using objective measures of vigilance (PVT; see Appendix A for statistics).

#### 3.1.2. MSL Performance

Behavioral analyses were performed on performance speed (mean time to complete a correct transition) and accuracy (% correct transitions) per block.

##### Speed

To assess improvement in performance speed across the training period, a repeated measures (RM) ANOVA was conducted on the 16 initial training blocks. This analysis showed a significant block effect (F_(15,345)_ = 17.39, *p* < 0.001) whereby performance speed improved across blocks of practice during initial training. The RM ANOVA conducted on the 4 post-training test blocks indicated no block effect (F_(3,69)_ = 2.48, *p* = 0.068), which suggests that performance became stable at post-training test (Figure 2A, upper panel, Training).

Similar analyses performed on the retest data showed that performance speed remained stable during both retest (F_(15,285)_ = 1.47, *p* = 0.116) and post-retest test (F_(3,57)_ = 1.04, *p* = 0.382; Figure 2A, upper panel, Retest). Significant online (t_(23)_ = 10.04, *p* < 0.001) and offline (t_(20)_ = 3.25, *p* = 0.004) gains in performance speed were observed (Figure 2B, upper panel). These results indicate that motor sequence learning was consolidated overnight.

##### Accuracy

Accuracy data were analyzed using a similar RM ANOVA as for performance speed. Results showed that accuracy remained stable across both training and retest sessions (block effect; training: F_(15,345)_ = 0.779, *p* = 0.701; post-training test: F_(3,69)_ = 2.026, *p* = 0.118; retest: F_(15,285)_ = 1.013, *p* = 0.442; post-retest test: F_(3,57)_ = 0.06, *p* = 0.98; Figure 2A, lower panel).

Significant online gains in performance accuracy were observed (t_(23)_ = −3.03, *p* = 0.006), but offline gains did not differ from zero (t(_20)_ = −1.78, *p* = 0.09; Figure 2B, lower panel). These results indicate that accuracy improved during initial motor sequence learning and was maintained overnight.

### 3.2. Functional Connectivity

EEG analyses were performed to test whether rsFC connectivity (i) changed after task practice and (ii) was correlated to motor performance, and - in particular - to initial motor sequence learning (online changes in performance) and overnight memory consolidation (offline changes in performance) processes. Brain-behavior correlations were only performed using measures of performance speed as performance accuracy remained rather stable during the experiment. Connectivity within the motor (MOT) network, and between the motor network and other networks (i.e., ventral and dorsal attentional VAN and DAN, language LAN, default mode DMN, and visual VIS) in the alpha, beta, and gamma frequency bands were considered of primary interest. Connectivity in (and between) non-motor networks and in other frequency bands (i.e., delta and theta) are reported for the sake of completeness but are considered exploratory. Results of exploratory analyses performed on networks of interest but in frequencies of no primary interest are reported in the main text. The results of all the other exploratory analyses are reported in the Appendix A.

#### 3.2.1. Training-Related Changes in Connectivity

A repeated measures ANOVA performed on rsFC metrics (z-scores) using session (pre- vs. post-task) as within-subject factor showed a significant decrease in connectivity in the gamma band within MOT (F_(1,20)_ = 4.53, p_uncorr_ = 0.046), and between MOT-VAN (F_(1,20)_ = 4.528, p_uncorr_ = 0.046) and MOT-LANG (F_(1,20)_ = 5.289, p_uncorr_ = 0.0324; Figure 3A within the black frame). Figure 3B depicts the significant decrease in gamma-band rsFC within the motor network. Similar results were observed in the gamma band for network pairs of no interest (see Figure 3A outside the black frame and Appendix A). No significant changes in connectivity were observed for the other frequency bands and network pairs of interest.

#### 3.2.2. Correlation between Connectivity and Online Gains in Performance

##### Pre-Task (Baseline) Connectivity

No correlations were found between rsFC prior to motor learning and subsequent online gains in performance in the networks and frequency bands of primary interest. Exploratory analyses did reveal correlations between pairs of networks of interest outside the frequency bands of interest. Specifically, in the theta band, connectivity between MOT and the other networks was negatively correlated with online gains in performance (Figure 4A within the green frame). Specifically, higher theta-band baseline connectivity between MOT-DMN (r = −0.51, p_uncorr_ = 0.032), MOT-DAN (r = −0.52, p_uncorr_ = 0.028), MOT-VAN (r = −0.51, p_uncorr_ = 0.031), and MOT-VIS (r = −0.54, p_uncorr_ = 0.020) was correlated with lower online gains in performance (Figure 4B). Similar results were observed in the delta band with negative correlations between online gains in performance and MOT-VAN (r = −0.47, p_uncorr_ = 0.049) as well as MOT-VIS (r = −0.52, p_uncorr_ = 0.026) between-network connectivity (see Appendix A). Negative brain-behavior correlations were also observed in network pairs not including MOT in the delta, beta, gamma, and theta bands (see Appendix A).

##### Training-Related Changes in Connectivity

Within the frequency bands and networks of interest, there were no significant correlations between inter-session changes in connectivity and online gains in performance (see Appendix A for exploratory analyses).

Altogether, correlation between connectivity in the network pairs of interest and online gains in performance indicate that higher baseline (pre-task) rsFC between the motor network and functionally distinct networks in the theta and delta frequency bands are related to poorer performance during initial motor sequence learning.

#### 3.2.3. Correlation between Connectivity and Offline Gains in Performance

##### Pre-Task (Baseline) Connectivity

In the beta band, a significant negative relationship was observed between offline gains in performance and pre-task connectivity between MOT-DAN (r = −0.52, p_uncorr_ = 0.049) and MOT-VIS (r = −0.66, p_uncorr_ = 0.008; Figure 5A,B), such that higher connectivity was related to poorer gains. This relationship was also significant in the alpha band between MOT-VIS (r = −0.56, p_uncorr_ = 0.029). Similar effects were observed between network pairs of no interest (see visual network outside the black frame in Figure 5A and Appendix A for details). Overall, these results indicate that higher baseline connectivity between the motor and other networks were related to poorer sleep-related motor memory consolidation.

##### Training-Related Changes in Connectivity

A significant positive correlation was observed between the learning-related changes in connectivity between MOT-VIS and offline gains in performance (r = 0.54, p_uncorr_ = 0.04) in the beta band (Figure 6), such that larger training-related increases in connectivity were related to better offline gains in performance. Similar results were observed in network pairs of no interest within the same band (see Figure 6A outside frame and Appendix A for details). These results demonstrate that training-related increases in beta-band connectivity are related to better sleep-related motor memory consolidation.

## 4. Discussion

In this study, we examined whether motor sequence learning influenced frequency-dependent functional connectivity of source-reconstructed hdEEG data, between and within previously described RS networks. Additionally, we tested whether these connectivity patterns were related to the motor learning and memory consolidation processes. Our data indicate that motor learning resulted in a decrease in gamma-band connectivity within the motor network as well as between the motor and ventral attention/language networks. Brain-behavior correlation analyses revealed that higher baseline RS connectivity between the motor and functionally distinct networks were related to both poorer learning (theta and delta bands), and poorer memory consolidation after sleep (beta band). Interestingly, learning-related increases in beta connectivity between the motor and visual networks were related to greater overnight gains in performance. Altogether, our results indicate that motor learning influences and is associated with specific patterns of frequency-dependent resting-state functional connectivity within the motor network, but also between the motor and functionally distinct networks.

### 4.1. Learning-Related Decreases in Gamma-Band Connectivity

Our EEG analyses showed that motor sequence task practice induced a decrease in gamma connectivity within the motor network, and between motor and ventral attention/language networks. Gamma oscillations are known to play an important role in motor processes [54]. Specifically, brief bursts of gamma activity occur at the movement onset in the primary motor cortex (M1) in an effect called gamma event-related synchronization [23,55,56,57]. Gamma power has also been reported to increase over parietal regions during early stages of motor learning and to progressively decrease as a function of practice [25,58]. Additionally, non-invasive brain stimulation studies have provided causal evidence for a link between gamma activity and motor behavior, such that gamma transcranial alternating current stimulation (tACS) to M1 can induce improvement in performance on a visuomotor task [59,60]. While the relationship between gamma *power* and motor processes is well described, the association between gamma *connectivity* and motor function is yet to be fully elucidated. Previous work has shown that gamma connectivity (as assessed with gamma-theta phase-amplitude coupling) is a crucial marker of memory formation [61,62]. Interestingly, decreases in gamma-theta phase synchronization over parietal regions have been reported during visuomotor learning [25]. These observations are in line with our findings, as we showed a significant decrease in gamma connectivity within the motor network, and between motor and distinct (ventral attentional and language) networks after motor sequence learning. Interestingly, both ventral attentional and language networks have been specifically related to gamma oscillations in previous work from our group [12], as nodes within these networks show preferential connectivity in the gamma band. Our findings not only indicate that the interaction between the (task-relevant) motor network and these distinct networks is tuned to the dominant connectivity frequency band of these networks [12], but also that these interactions decrease after learning. We speculate that the decrease in gamma-band connectivity—between the motor and attentional networks in particular—reflects a lower need for control and attentional processes after the task is well learned [63]. It is worth noting however, that the gamma connectivity results discussed above did not survive correction for multiple comparisons and must therefore be considered with caution.

### 4.2. Baseline Theta, Delta, and Beta Connectivity Predicts the Subsequent Learning and Consolidation Processes

Brain-behavior correlation analyses indicated a negative relationship between performance and baseline connectivity in the beta, delta and theta bands, between the motor and functionally distinct networks including the visual, attentional, and default mode networks. Specifically, higher baseline delta and theta connectivity were related to poorer initial motor learning, and higher baseline beta connectivity predicted lower overnight consolidation. Research into the functional significance of *theta*-*band* oscillations has highlighted an integral role of this rhythm in memory formation [64,65] and retrieval [66,67] processes, which appear to be supported by gamma-theta cross frequency coupling [68,69]. In the motor domain, theta power over the right parietal region has been positively associated with the degree of learning and retention on a visuomotor task, and gamma-theta phase coherence has been shown to be critical for movement planning [25]. With respect to *delta* oscillations, previous research has more commonly associated delta rhythm to sleep rather than wake states (see [70]), where they play an important role in memory consolidation processes [71] via hippocampal-cortical [72] and cross-cortical [73] communication. Similar as above, research into the relationship between delta and theta connectivity and motor behavior is rather scarce. Our data therefore provide the first evidence for a link between theta and delta connectivity at baseline, and subsequent motor learning. In contrast, the role of *beta* oscillations in motor function is well established, as beta-band desynchronization over sensorimotor areas is a well-known marker that precedes voluntary movement [74]. Of particular interest for the present research, recent MEG studies have shown that pre-task beta-band connectivity was negatively related to subsequent motor learning [16,17]. Our results extend these previous findings as they show that baseline beta connectivity between the motor network and the visual and attentional networks were related to poorer overnight gains in performance. Taken together, our results indicate that greater baseline connectivity across several frequency bands (delta, beta, and theta) within the motor network and between the motor and distinct networks is related to poorer learning and consolidation processes. These findings corroborate evidence, gathered from various neuroimaging techniques [10,17,75], that greater network segregation at rest (i.e., lower large-scale resting-state connectivity) is beneficial for motor behavior. Our research confirms these previous observations, and reveals that network segregation is not only beneficial for subsequent motor learning, but also for the sleep-related consolidation process.

### 4.3. Training-Related Increases in Beta Connectivity Are Related to Better Consolidation

Lastly, our brain-behavior analyses also highlighted a positive relationship between motor memory consolidation and training-related *changes* in beta-band connectivity between networks. Specifically, greater increases in beta connectivity after task practice between the motor and visual networks were related to better consolidation after a night of sleep. While these results are based on a lower sample size and did not survive FDR correction for multiple comparisons, they are partly in line with previous research. Indeed, a recent MEG study showed a positive correlation between (early) motor memory consolidation and training-related changes in rsFC between the primary sensorimotor area and distant seed regions (frontal and parietal areas in this research) [17]. It was argued that such connectivity patterns support attentional processes that are beneficial for the subsequent consolidation process [17]. It remains unclear whether such interpretation can be extended to our findings (motor-visual increase in connectivity) as visual processing is rather minimal in our motor task. Nonetheless, our previous work has shown that task representations including more abstract (spatial) features of the task—that partly rely on visual processing—can develop during initial learning [76]. Based on evidence that such task representation preferentially consolidates during sleep [76], we suggest that increased motor-visual beta connectivity might support such a task component, and predict the subsequent sleep-related consolidation process. This interpretation is speculative however, and future research is warranted to elucidate the functional significance of such connectivity patterns. Interestingly, in addition to the correlation between consolidation and motor-visual beta connectivity *changes* discussed above, our results also showed that lower *baseline* beta connectivity between these networks was predictive of greater consolidation. An exploratory correlation analysis between baseline and training-related changes in motor-visual beta-band connectivity indicated that the lower the baseline connectivity, the greater the training-related changes in connectivity (r = −0.535, *p* = 0.04). Altogether, it is therefore tempting to speculate that higher baseline network segregation (i.e., lower baseline connectivity) might allow greater experience-induced increases in connectivity that might, in turn, favor the sleep-related consolidation process. However, this interpretation is hypothetical and warrants further investigation.

## 5. Conclusions

In the present high-density electroencephalography study, we investigated how frequency-dependent functional connectivity in resting-state networks is modulated by and related to motor learning and memory consolidation. Our results show that connectivity —in various frequency bands including beta, delta, gamma, and theta bands—between the motor (task-relevant) and functionally distinct networks is critically linked to motor learning and memory consolidation. These findings expand on past research consistently linking beta-band connectivity to motor learning processes. They also provide evidence that the study of frequency-dependent functional connectivity in large-scale resting-state networks brings additional insight into motor function.

## Figures and Tables

**Figure 1 brainsci-12-00530-f001:**
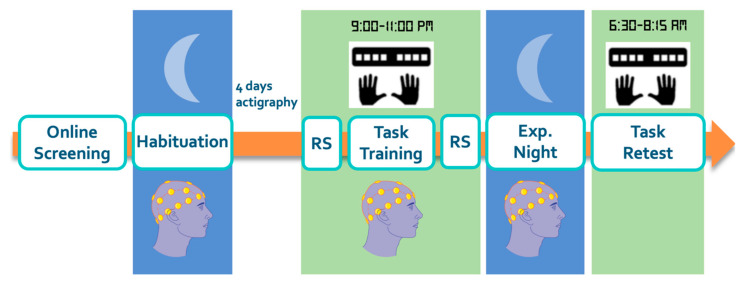
After completing online questionnaire screening, participants came into the lab for a habituation night with the hdEEG system. On the experimental night, the motor sequence learning task was completed before sleeping and 45 min after waking the next day. Resting-state hdEEG data were acquired before and after the pre-sleep task training. Sleep of the experimental night was also monitored with hdEEG. Participants wore an Actiwatch and completed a sleep diary for the four days prior to the experimental night to confirm adherence to a regular sleep schedule. EEG illustration from smart.servier.com (accessed on 1 February 2022).

**Figure 2 brainsci-12-00530-f002:**
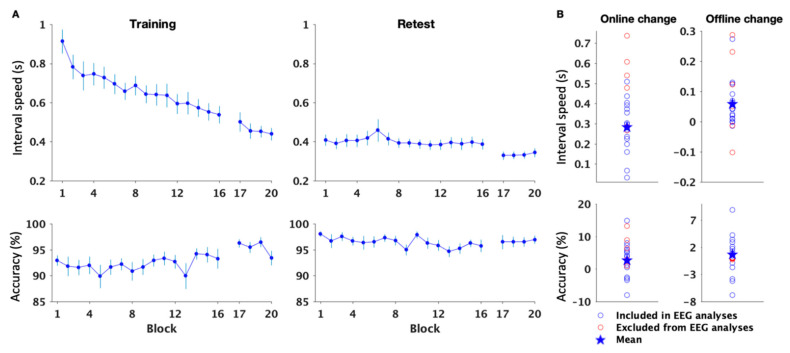
(**A**) Performance speed (inter-key interval in s, upper panel) improved across the training session (*n* = 24) and showed stable levels during the post-sleep retest (*n* = 20). Accuracy (% correct transitions, lower panel) remained stable across the training and retest sessions. Bars represent SEM; (**B**) Significant online gains in performance (*n* = 24) were observed for both speed and accuracy (upper and lower panels, respectively). Offline gains in performance (*n* = 21) were significant for the speed but not the accuracy measure. Each open circle represents a participant. The star represents the group mean. Red circles depict participants that were excluded from EEG analyses due to EEG data quality issues (see methods).

**Figure 3 brainsci-12-00530-f003:**
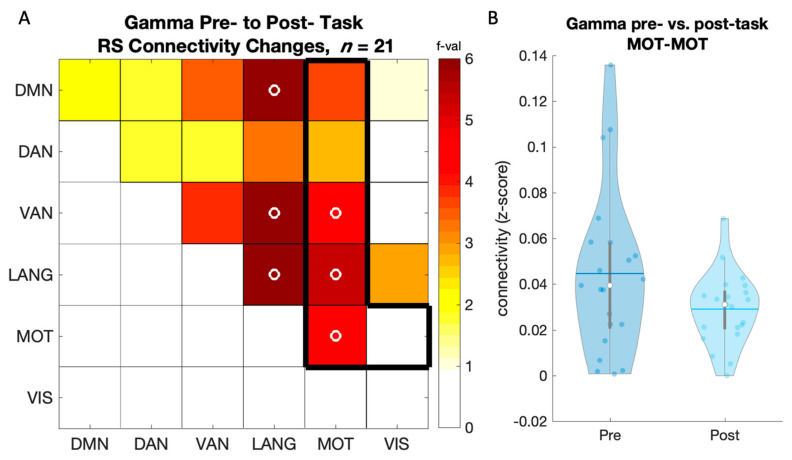
(**A**) Pre to post-learning changes in RS gamma-band connectivity. The black frame highlights the network comparisons of interest. Color scale displays ANOVA F-values. Open circles (o) indicate significant results at *p* < 0.05 uncorrected for multiple comparisons. Note that none of these results survived FDR correction; (**B**) rsFC within the motor network in the gamma band decreased from pre- and to post-learning (*n* = 21). Colored circles represent individual data, jittered on the *x*-axis. Black horizontal lines represent means; white circles represent medians. Violin plots were created with [53].

**Figure 4 brainsci-12-00530-f004:**
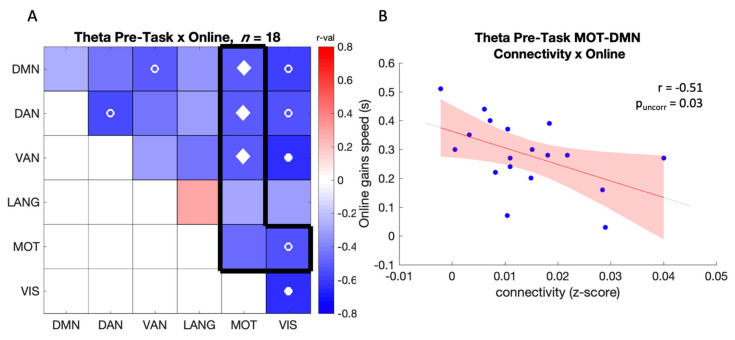
(**A**) Correlation between online gains in performance speed and pre-task connectivity in the theta band. The black frame highlights the network comparisons of primary interest. The color scale represents r-values. Open circles (o) indicate a significant correlation at *p* < 0.05 uncorrected, (•) *p* < 0.05 FDR corrected across all 21 comparisons, and (♦) *p* < 0.05 FDR corrected across the 6 comparisons of pairs of interest (see black frame); (**B**) Scatter plot showing the negative relationship between online gains in performance speed (s) and pre-task MOT-DAN connectivity in the theta band (*n* = 18).

**Figure 5 brainsci-12-00530-f005:**
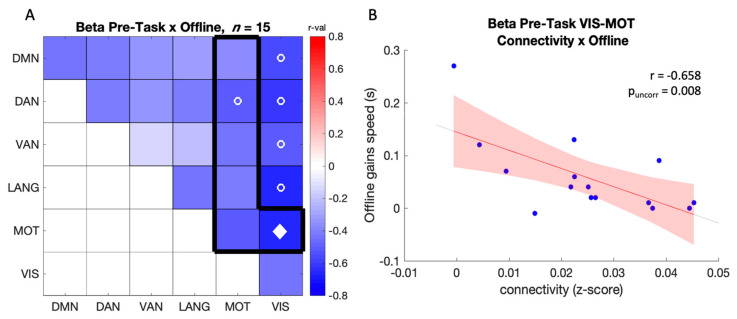
(**A**) Correlation between offline gains in performance speed and pre-task connectivity in the beta band (*n* = 15). The black frame highlights the network comparisons of interest. The color scale represents r-values. Open circles (o) indicate a significant correlation at *p* < 0.05 uncorrected and (♦) *p* < 0.05 FDR corrected across the 6 comparisons of pairs of interest (see black frame); (**B**) Scatter plot showing the negative relationship between offline gains in performance speed and pre-task connectivity between MOT-VIS in the beta band (*n* = 15).

**Figure 6 brainsci-12-00530-f006:**
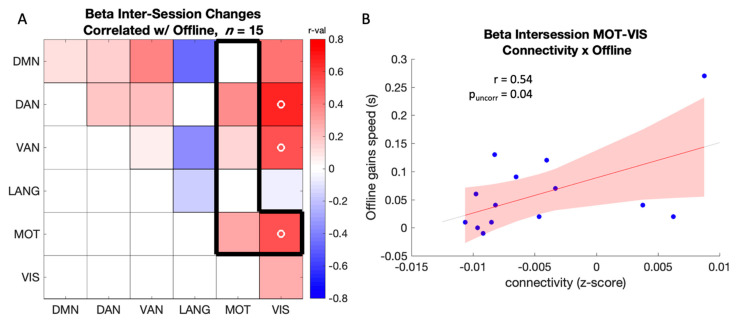
(**A**) Correlation between offline gains in performance speed and intersession changes (POST-PRE) in connectivity in the beta band. The color scale represents r-values, and the black frame highlights the network comparisons of interest. Open circles (o) indicate a significant correlation at *p* < 0.05 uncorrected. Note that none of these results survived FDR correction; (**B**) Scatter plot showing the positive relationship between offline gains in performance speed and intersession increases in beta-band connectivity between motor and visual networks (*n* = 15; note that 2 data points are overlapping in the scatter plot).

**Table 1 brainsci-12-00530-t001:** Sample size of participants per analysis.

Behavioral Analyses	Sample Size
Task Training	*n* = 24
Task Retest	*n* = 20
**EEG Analyses**	
Pre-Post	*n =* 21
Correlation with online gains	*n* = 18
Correlations with offline gains	*n* = 15

**Table 2 brainsci-12-00530-t002:** MNI coordinates of rsFC ROIs, which were derived from fMRI rsFC literature.

Seed	MNI Coordinates			
	X	Y	Z	
lANG	−57	−63	17	DMN
rANG	56	−63	18	
PCC	5	−58	29	
MPFC	−5	35	−9	
lIPS	−27	−61	50	DAN
rIPS	26	−60	48	
lFEF	−30	−9	52	
rFEF	30	−9	55	
rTPJ	60	−43	16	VAN
rIFG	42	5	1	
lTPJ	−54	−33	−4	LANG
lIFG	−47	14	1	
lSMA	−1	−17	55	MOT
lCS	−45	−17	49	
rCS	45	−17	49	
lS2	−42	−13	10	
rS2	42	−13	10	
lV1V2	−27	−81	−13	VIS
rV1V2	27	−81	−13	
lMT	−45	−81	4	
rMT	45	−81	4	

## Data Availability

The ethical approval granted by the local ethics committee does not permit the publication of data online.

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
