# Peer review of "Connectivity in Large-Scale Resting-State Brain Networks Is Related to Motor Learning: A High-Density EEG Study"

_brainsci, 2022, doi:10.3390/brainsci12050530_

Round 1

Reviewer 1 Report

The paper is well organized with proper structure. The bibliography is sufficient and well given.

Specifically, the technical terms are explained in detail and the topic of the paper is clear and understandable.

The presented methodology and the results are clearly communicated, with the necessary background for the readers included in the paper.

The review of the state-of-the-art is sufficient. It includes references to other relevant studies that have been previously proposed for the discovery of relations.

The novel contribution of the paper is highlighted, as well.

The conclusion section includes a discussion about the results obtained by this work, but it doesn't demonstrate previous works on the analysis of the same or similar data.

Reviewer 2 Report

This study used detailed EEG methodology to investigate brain functional connectivity that is associated with a motor learning task. The research design was robust, measuring behavioural performance at baseline and after they were trained on the task, including a period of overnight memory consolidation.  Functional connectivity was measured broadly by correlation with and between well-established resting state functional networks, and thus was not restricted only to networks associated with motor control.  

The use of terminology was confusing and made it difficult for me to link the sections of the discussion with the experimental findings. For example in the discussion they state that “we tested whether these connectivity patterns were related to the motor learning and memory consolidation processes.” However, I could not find the term “memory consolidation” in any of the results section or in any of the figures. Instead the figures refer to “Online” and Offline” effects. It would helpful if the authors ensure that the terminology and technical term are consistent and aligned throughout the manuscript.

Section 2.1 Participants

This section is very difficult to follow. This should be supplemented by a table clearly showing how many participants were included for each of the experimental comparisons. It is also important to know whether this study had sufficient statistical power to detect the effects under investigation, once all the problematic participants were excluded.

There appears to be a mismatch between the numbers of participant reported and some of the data points in the graphs, including Figure 3B, 4B & 5B. The slopes and p values of the regression lines should be given for Figures 4B & 5B.

The legend of Figure 4 states that FDR was corrected across the 6 comparisons of interest. Why was FDR not corrected for all the statistical comparisons?

Reviewer 3 Report

This manuscript presents an interesting and novel approach to explore relationship between network connectivity and motor learning and consolidation using EEG based network connectivity analysis. EEG data is recorded during simple motor task and the data collection is repeated the following morning of the experiment. Connectivity between sensorimotor network and other well defined networks are examined.  Results do not show change training related changes in network connectivity with FDR correction. Results show trend of correlation between theta band  baseline connectivity between sensorimotor and other networks and online gains in speed. The correlation is not significant with correction. Beta band connectivity connectivity of sensorimotor and visual network correlated with less offline gains in speed. Overall the study was well designed but there are too many outcomes and too many analyses that is hard to understand the main conclusion of this paper. The supplementary material attachment also include the data during sleep, which is not included in the main paper. I do not understand the importance of EEG data during sleep in this paper and how it was used to examine the main hypotheses. 

I suggest the following to improve the paper. 

The discussion can include subsections that highlight each of the main findings. The list of ROIs is included in the supplementary material. It is important to include the list in the paper and justify how those ROIs has been selected. Finally if the data during sleep are not used in the main analyses, may be they can mentioned but not included in the paper or the supplement. 
